# Advances in Flipped Classrooms for Teaching and Learning Forensic Geology

**Roberta Somma** 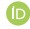

Department of Mathematical and Computer Sciences, Physical Sciences and Earth Sciences, University of Messina, 98166 Messina, Italy; rsomma@unime.it

**Abstract:** One of the most famous criminal investigations involved the use of forensic geology to assist law enforcement agencies in the homicide case of the Italian honorable Aldo Moro. Notwithstanding this important tribute to forensic geology, in Italy, the role and value of using geological and soil materials (known as earth materials) to support law enforcement agencies in solving criminal investigations remain uncommon. This absence may be due to few educational courses for geology undergraduates/graduates devoted to laboratory and field training in forensic geology. The flipped classroom model may encourage a modern educational approach for teaching and learning forensic geology. The designed flipped classroom model applies theoretical concepts for forensic geology, which is learned by the attendees at home, whereas the class activities are devoted to laboratory and field experiences assisted by teaching staff. The laboratory activities involve techniques for collecting geological trace evidence and comparing color/sedimentological/mineralogical/microfossil features, whereas the field experiences consist of sampling strategies, search activities for burials, and field surveying. This approach has been trialed by the Messina University since 2014 and represents a successful tool for multitasking teaching and learning aimed to further develop forensic geology, encourage the inclusion of forensic geologists within the police enforcement in Italy, and improve the knowledge of law experts such as prosecutors and defense lawyers.

**Keywords:** flipped classroom; academic education; science education; forensic geology education; experiential; interactive; collaborative; fieldwork; training; teaching and learning strategies

## 1. Introduction

Forensic geology is the discipline that applies the scientific principles and techniques of geosciences to solve criminal cases [1–12], assisting law enforcement and the judicial system, reconstructing events, or providing evidence which are potentially used in a court of law [12]. The geoforensics applications [10] may concern human rights violations, counterterrorism, war crimes, environment, and property. Most of the crimes are related to homicides, corpse concealments, hit and run incidents, kidnappings, sexual assaults, animal maltreating and wildlife crimes, stone-throwings, robberies, thefts in apartments, vandalism, fraud and financial crimes, geotechnical and geohazard problematics, and environmental damages [8,9,11,13–27].

Forensic geologists use geological and geochemical evidence associated with minerals, soils, sediments, anthropogenic materials, microfossils, rocks, groundwaters/surface waters, and gases to solve criminal cases [12,27]. A wide range of info-investigative data may derive from the study of geological trace evidence to reconstruct "historical facts" starting from applying scientific laws and technical-scientific methods on which the discipline is based. This information is required so that informed decisions can be provided to support and help police, law enforcement and forensic scientists with criminal investigations [27]. Criminal cases relating to murders or suspicious deaths occurring in open places significantly differ from those occurring indoors due to the high possibility that inorganic traces or micro-traces may be transferred from the crime scene to items linked

with an offender [28,29]. Anthropogenic and inorganic materials may generally provide robust physical evidence. They may transfer (Locard's principle; Figure 1) from the crime scene, allowing to associate or exclude a suspect with or from the crime scene or a suspect or alibi area.

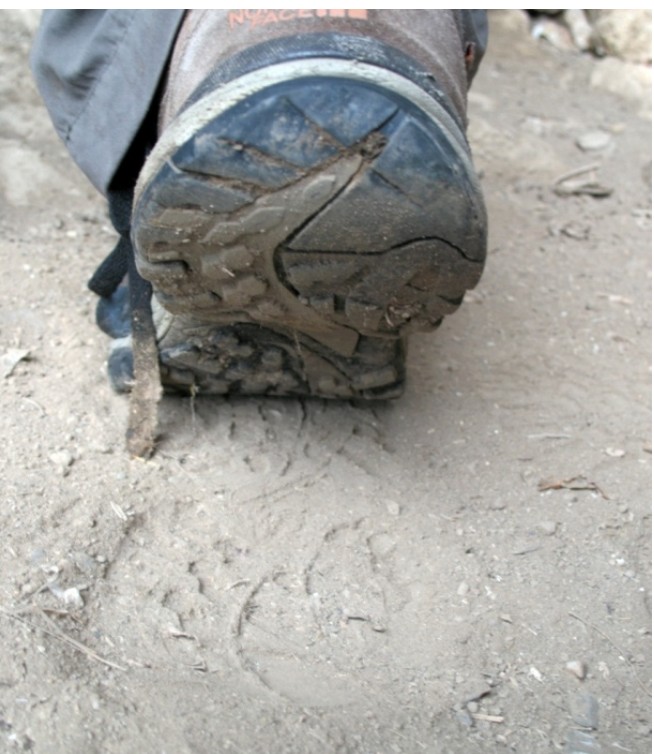

**Figure 1.** Example of the Locard's exchange principle in forensic geology. The contact between the earth's surface and the shoe allows the soil's transfer from the topsoil to the shoe sole because of the exchange of soil material. Source: author.

The primary purpose of forensic geology trace evidence is to associate (or exclude) samples of inorganic and organic materials found on an item (such as human remains, shoes, clothing, shovel, motor vehicle, where the geological physical evidence is of unknown provenance) with soils/sediments of a specific location of known origin. Forensic geology applications may establish whether a questioned or unknown sample is comparable (compatible or similar) to a known sample for compatibility/exclusion and provenance purposes [4,9,11]. The comparative analyses of geological traces/micro-traces with inorganic materials found at the crime scene or areas of investigative interest may provide fundamental info-investigative data helpful in establishing:

- The pre-mortem presence of the victim or suspect on a crime scene;
- The walking carried out by the victim or suspect on a site;
- If there was a transfer of the victim's body in places different from the primary crime scene;
- In some forensic cases, also the cause and manner of death assisting medical examiners and coroners.

References to forensic geology over the last 20 years are equal to a mean value of 23,000/y (Figure 2A); it is possible to observe that the number of publications (Figure 2B) and academic meetings/events in geosciences with special sessions devoted to forensic geology is strongly increasing [12], enforcing its position among other forensic sciences disciplines. A strong tradition of scientific research associated with the topic of forensic geology has developed among the academic communities and experts, mainly in the USA [1,2,4,5,30–32], UK [6–10,12,17–19,33–41], and Australia [20,27].

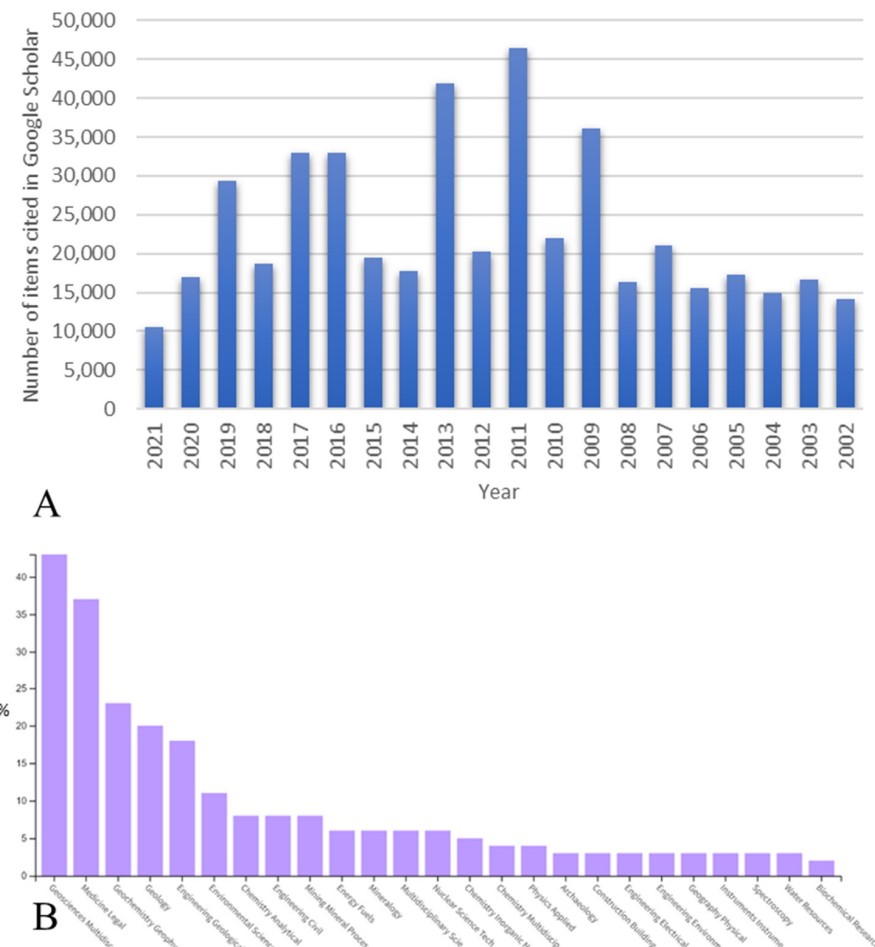

**Figure 2.** (**A**) Forensic geology has been cited about 23,000/y in the last 20 years. (**B**) Publications (N. 191) selected from Web of Science Core Collection for the 1989–2022 time interval. The most numerous papers on forensic geology are produced in the USA (N. 61), UK (N. 27), Canada (N. 20), and Australia-Germany-China (N. 14). The main topic concerns the Geosciences disciplines (N. 43). Source: own elaborations based on Google Scholar and Web of Science Core Collection [42,43].

One of the most famous criminal investigations in the world, which is closely associated with the use forensic geology, involves the murder case of the honorable Aldo Moro, kidnapped and killed by terrorists in 1978 in Rome (Italy). Geologist Prof Gianni Lombardi analyzed the geological physical evidence found in the trousers and moccasin soles of Aldo Moro (about 1 g) and compared them with less than one hundred samples of beach sands collected along a 150 km long sector of the Lazio coast. Sedimentological and mineralogical comparative analyses established that the provenance of the geological traces could be derived from the sector of the Lazio coast, stretched from Palidoro to Focene, to the North of the Fiumicino airport of Rome [44]. This significant result contributed to the historical reconstruction of the case, suggesting that Moro could have walked in that specific area only a few days before being killed [44]. This reconstruction, considering the recent discovery of the terrorists' message N. 13 divulgated by mass media [45], could be compatible with a failed terrorist plan to transfer Aldo Moro to Genoa [46]. Notwithstanding the importance of Lombardi's significant contribution to forensic geology, the role and value of using geological and soil materials (known as earth materials) to support law enforcement agencies in solving criminal investigations remains uncommon in Italy. This absence may be associated with the lack of knowledge in this earth sciences sector because specific university and police courses are few. Furthermore, regarding legal medicine aspects, it is highlighted the difficulty of preserving geological physical evidence during the removal of

a corpse from the crime scene. These fundamental issues make the possibility that traces and micro-traces of geological materials may be contaminated or completely dispersed, especially if not recognized by skilled experts.

The forensic geology theoretical principles at the base of this complex discipline, cover all the fields of earth sciences. Based on the above information, the most common activities and subject matters faced by the forensic geologists may be synthesized in:

- Crime scene examination [12];
- Forensic soils and traces/micro-traces comparisons [4,9,11,12,18,27,34,46];
- Provenance and source to sink studies on geological physical evidence [47];
- Ground search for clandestine burials (corpses, weapons, money, narcotics, fugitive bunkers, etc.,) [12,36,37,48–52];
- Environmental forensics [12,22,25,53–55].

Different analytical methods are applied to geological physical evidence to identify, characterize, and quantify their inorganic, organic, and anthropogenic components (colorimetric, sedimentological, mineralogical, paleontological, physical, chemical, and biological analyses). These analyses aim to obtain information on the compatibility degree among unknown and known samples and the possible provenance of geological trace evidence. The estimate of the compatibility degree needs a variety of factors to be quantitatively and qualitatively compared [11]. Fitzpatrick et al. [20,27] apply a scale of values to determine the "degree of comparability" between questioned and known control soil/geological samples but Morgan and Bull [56] and others consider this approach too restricting. Laboratory analyses may be non-destructive (the sample maintains its original feature) or destructive (the sample modifies its features and or is destroyed). Where possible, it is preferable to apply non-destructive methods and to preserve a quote of the original sample for further analyses.

Geologists applying forensic geology need to master sedimentology, physical geology (mineralogy and petrology), micropaleontology, geochemistry, and soil sciences to compare geological physical evidence. To search for buried items, it is necessary to have a solid knowledge of stratigraphy, sedimentary petrography, geomorphology, regional geology, remote sensing, GIS, applied geophysics, and soil sciences. One of the most complex application of geoforensics deals with environmental crimes; in these investigations, geologists need to be knowledgeable in stratigraphy, sedimentary petrography, igneous and metamorphic petrography/petrology, hydrogeology, geochemistry, structural geology, and applied geology and geophysics. Moreover, each forensic geologist/soil scientist must be confident in criminal procedure to establish methods and techniques correctly [27]. In each of these different cases, forensic geologists must have expertise not only with all the disciplines mentioned above but also they need to be well trained in sampling techniques, and field experience aimed to answer questions posed by the prosecutor or the defense lawyer, according to ethical principles and complying to protocols and procedures or best practice manuals [57] developed by the scientific community.

Training in the field and laboratory for an expert in forensic geology is crucial. Carlos Martín Molina Gallego organized a forensic geology training course in Mexico in 2010, and in the same year, another workshop was organized in Portugal. In 2012, the International Union of Geological Sciences (IUGS) Initiative on Forensic Geology (IFG) organized and ran a two day training course on search for Police forces at the Queensland Police Training Centre in Australia. Currently few opportunities exist for academic or law enforcement officers to attend certified forensic geology search and trace evidence courses. Based on the above information, the expertise that a geologist has to possess to become a skilled expert in forensic geology is remarkable and difficult to acquire without having the opportunity to improve his/her knowledge on these complex interdisciplinary topics.

Experiential and interdisciplinary pedagogy in forensic geology could be helpful in achieving and improve learning outcomes. It is desirable to enhance careers in geology undergraduate to postgraduate courses to train attendees to undertake crime scene investigation correctly. Simulated fields and laboratory activities on forensic geology may

represent non-traditional and experience-based learning of paramount importance for education on this topic to enhance the learning outcomes. The present paper provides a modern flipped classroom-based educational approach addressed to geology undergraduates/graduates and teaching staff for teaching and learning fundamental principles, methods, and practices required of skilled forensic geologists. The model may represent successful tools for multitasking teaching and learning strategies necessary to further develop forensic geology, especially in Italy, to encourage the inclusion of forensic geologists within police enforcement and improve training activities for experts of the prosecutor and the defense lawyers.

## 2. Materials and Methods

Teaching depends on the issue's knowledge and the use of suitable and modern pedagogical approaches to facilitate the learning process of the learners. Modern teaching methods for geology [58] and engineering undergraduates [59] demonstrate that experiential work-integrated learning may improve students' skills. The flipped or inverted classroom is a modern student-centered teaching approach that overturns the traditional learning cycle [60–65]. This pedagogical learning method foresees that the conventional homework is done at school/university, whereas the classic school/university work is done at home, promoting learning democratization [66]. Students attending flipped classrooms become active learners [67], whereas the teachers become face-to-face guides during the practice, maintaining a facilitator role. The flipped classroom method has been employed for teaching scientific disciplines in university courses worldwide [68].

A modern and valid teaching and learning approach for the forensic geology discipline may be represented solely by the flipped classroom method. Several such approaches are used in the American popular television series (CSI, Bones, NCSI), television programs, and documentary movies or radio programs (CSI Milan by radio105) devoted to forensic sciences where it is expected that traces of mineral grains seen under a microscope or clandestine burials of disappeared people assume a role during the criminal investigation fiction. This phenomenon increased forensic studies in higher education and university in the UK [69,70].

The flipped "classroom" model for teaching forensic geology is designed for geology undergraduates or postgraduates who desire to familiarize themselves with applied geoforensics. In the proposed flipped classroom model, documentaries, fiction, short video lectures, peer-assisted learning, podcasts, and scientific articles/books offered by the teaching staff may be viewed by the students/attendees at their homes before the class session. Classwork may be devoted entirely to exercises and laboratory or field experiences for collaborative activities. Students are involved in field and laboratory environments, guided by the teaching staff. The primary references used are reported in the present paper.

The learning method aims to stimulate the students' curiosity and interest in forensic geology and enhance the learning of its principles and practices during homework before approaching inclusive and dynamic activities at school/university. The understanding of how to manage the data set, performing forensic geological physical evidence comparisons, recognizing evidence of pollution in the groundwater/underground, and searching for concealed items in the ground, accompanied by the development of critical thinking and professional ethic, are the main goals of the current teaching and learning approach applied to forensic geology. On the basis of the above, the Messina University, since 2014, has organized courses on forensic geology with field training and laboratory experiences for attendees from all over the world. The first field training and laboratory experiences were organized by the Messina University on the scientific campus during an international workshop held for the students of the post-degree master's course in forensic geology and police officers (Figure 3). In the following years, experts involved in the teaching staff were forensic archaeologists, anthropologists, botanists, chemists, coroners, engineers, entomologists, experts in search with detection dogs, geologists, and physics, most of them being closely associated with police forces.

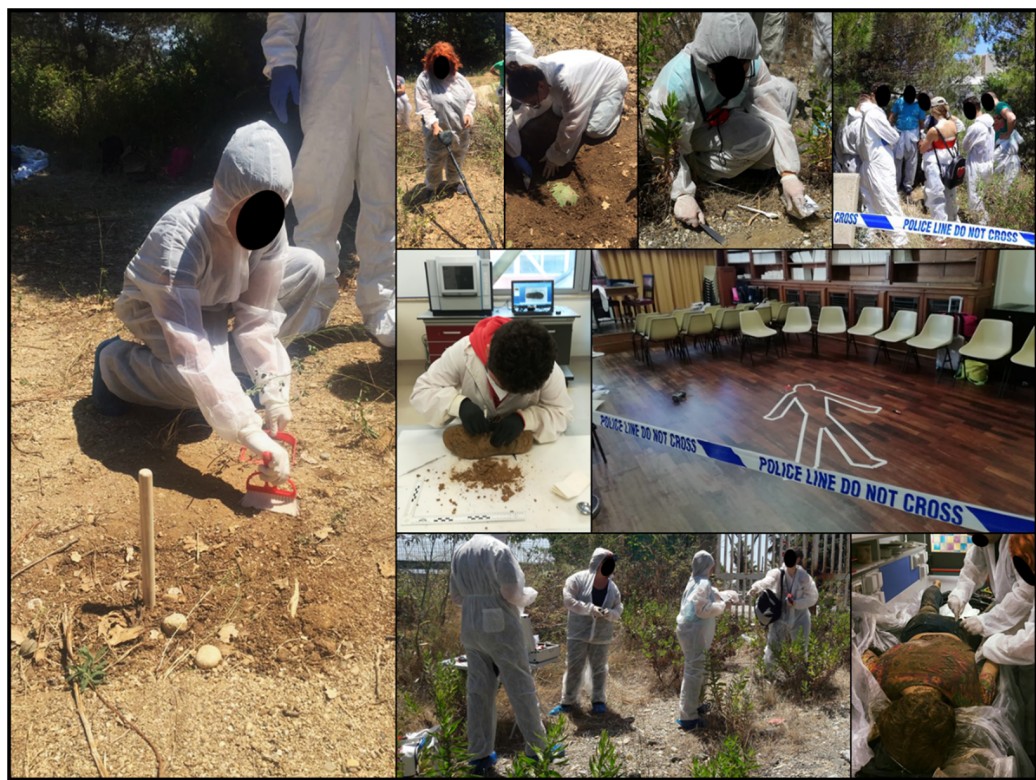

**Figure 3.** Photographs of attendees during the forensic geology training activities organized by the Messina University since 2014. Source: own development.

Field and laboratory experiences require that attendees apply geological physical evidence and conduct laboratory and field instruments; they are guided face-to-face by the teaching staff through a series of instructions and questions to obtain familiarity and improve their specific knowledge. The teaching staff stimulates attendees to cooperate in working groups and discuss together comparing different critical opinions on obtained results.

The planned field and laboratory experiences are organized to facilitate the learning of scientific methods and techniques to be applied for the following forensic activities:

- Analysis of the crime scene and techniques for geological physical evidence collecting and analyses;
- Ground search for simulated clandestine burials;
- Environmental forensics applied to landfills and related water pollution related to landfill leachate.

Experimental research on the decomposition phenomena and the ground transformations using human corpses is not admitted in Italy in contrast to the USA (The Body Farm, Tennessee); consequently, different experiences were organized by the Messina University in simulated and predisposed fields using concealed items such as plastic mannequins, steel knives, and metal boxes. Simulated crime scenes for the searching activities and the geological physical evidence sampling [71] were prepared in two different areas of the Peloritani Mountains (NE Sicily, Southern Italy). The first site is localized on the slope surrounding the scientific campus of the Messina city (Lat. 38°15′38.61″ N, Long. 15°35′48.65″ E, Elevation 70 m a.s.l.). The substrate is composed of Quaternary siliciclastic sedimentary rocks (the Pleistocene gravels and sands of the Messina Formation [72–74]) overlain by topsoil with a thin litter layer where a pine and oak wood is developed [75]. In the soil on the campus crime scene, the holes for the concealment were dug by pick and shovel, being a difficult diggable soil [76]. The second site is in the Alì village, along the Ionian slope (Lat. 38°1′44.19″ N, Long. 15°25′27.00″ E, Elevation 524 m a.s.l.). The substrate is formed

of a Paleozoic metamorphic basement (phyllites of the Mandanici Unit [77–80]) overlain by a cultivated land with olive trees [75]. In this case, the dig was made by a mechanical excavator in a cultivated soil (hole with dimensions 2 × 1 m and 1 m of deepness [75]) to offer a different experimental context.

Field skills relating to environmental crimes are learned during the field-trip accessing the site of a closed landfill and visiting the surrounding areas of another closed landfill. In the first case, the substrate of the landfill site is formed of siliciclastic sedimentary rocks (Tortonian gravels of the San Pier Niceto Formation [81–83]) in the Ionian slope of the Peloritani Mountains. In the second case, the investigated area, localized in the northern sector of the Peloritani Mountains, is out of the landfill extending along streams crosscut by normal faults affecting the Paleozoic metamorphic rocks (Augen gneiss of the Aspromonte Unit [78,84]) and the Pleistocene Messina Formation.

Applied methods, techniques, activities, and used portable instrumentations in the fieldwork consist of:

- Sampling techniques and chain of custody;
- Metal detector;
- Electrical Resistivity Tomography;
- Seismic Tomography;
- GPR;
- Thermography;
- Laser scanner;
- Drone;
- Stratigraphic and archaeological digging.

Sampled materials and investigated matrices are soils, rocks, and underground water. Data and information provided to the attendees are related both to closed criminal cases and the author's scientific research experiences.

The most common laboratory activities concerning forensic geology consist of techniques for collecting geological physical traces/micro-traces adhering to questioned items and comparing main geological features (color, texture, mineral composition, etc.,). Applied methods, activities, techniques, and used instrumentations in the laboratory work consisted of:

- Sample collecting techniques and sample preparation;
- Color determination (Munsell Soil color book with color charts);
- Sedimentological, micropaleontological, and mineralogical determination (optical microscopy, XRD diffraction, SEM-EDS, spectroscopy, XRF).

Each attendee is provided with a geological tool kit, which includes personal protective equipment (forensic suits, overshoes, gloves, facemasks, glasses).

## 3. Results

It is assumed that all attendees have a geological background, the teaching staff invites students to study at home (before the training) the theoretical and fundamental concepts/principles of forensic geology to learn the value of geological physical and geochemical evidence. This session focuses on:

- Forensic science, criminalistics vs. criminology;
- Forensic geology applied to criminal investigations;
- Unknown and known samples and collection techniques in the field;
- Geological physical evidence collection techniques in the laboratory;
- Preparation of the sample;
- Grain size separation;
- Color determination;
- Sedimentological analyses;
- Mineralogical and petrographic analyses;
- Chemical/Geochemical and physical analyses;

- Microfossil analyses;
- Search method for concealed items/targets;
- Search method for pollution of underground waters.

The successive stage of the field and laboratory training is inclusive and organized in groups of attendees to whom the teaching staff poses some questions and indicates the activities to carry out with their assistance. At the end of all the activities, each group has to edit a synthetic technical and scientific report centered on the expert opinion.

### 3.1. Field Training

Messina University generally offers the training during a summer school of 40 h and a geology master's degree course in forensic geology of 42 h. Field training is mainly devoted to:

- Sampling strategies and geological physical evidence collecting in the simulated crime scenes and alibi sites for control samples;
- The ground search for buried targets through surface observation and subsurface anomalies detection;
- The preliminary survey of a site hosting a landfill to search for pollution evidence.

Outdoor activities may be enhanced by creative experiences such as those offered to the attendees in 2017 when field experience foresaw a judicial inspection for a simulated homicide involving a professional actor playing the serial killer's role, whereas other persons the roles of prosecutor, police agents, and CSI investigators (Figure 3).

### 3.1.1. Field Training in Sampling Strategies and Geological Physical Evidence Collecting

The teaching staff poses the following question to the attendee groups: *Did the suspect walk on the crime scene and the alibi area?* The staff reminds the attendees to complete the training with a second activity in the laboratory to answer this question. The team communicates that for the exercise, they may use: cords, wood pickets, evidence flags and markers, scale bars, rulers, evidence bags, indelible pens, trowels, steel spatulas, small shovels, spoons, brushes, toothbrushes, topographic maps, aerial imagery, GPS, and the photographic camera. The attendees choose the best sampling strategy, considering other related disciplinary areas such as geomorphology and the degree soil homogeneity. Then they subdivide the spaces into a square grid made up of cord. Each attendee group, after having taken photographs of each site of sampling (with a scale bar, N direction, and evidence marker), and geolocalized the sample using a GPS device, samples the geological physical evidence (about 200–300 g [75] or a volume of 50 mL [11]) from the shallow topsoil $30 \times 30$ cm wide, collecting the samples in plastic, linen, or paper bags. Collecting wet or moist soils in "plastic bags" is not recommended as the soils can alter with time: i.e., the soils become "reduced" in that iron oxide minerals such as hematite and goethite can alter/transform to "ferrous iron minerals" and change color to grey or gleyed colors. As a consequence, it is best to collect samples in strong "linen bags or paper bags" or collect in plastic bags but dry the soils within hours after being collected (Figure 4). Attendees learn to correctly manage the packaging and transporting techniques from the field to the laboratory and compile the chain of custody documentation.

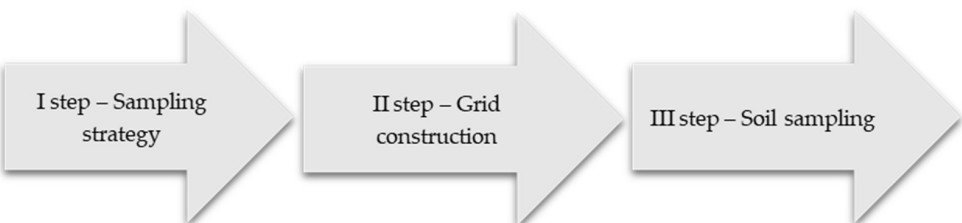

**Figure 4.** Workflow of the activities. Source: author.

### 3.1.2. Training in a Ground Search for Buried Targets

Both simulated fields on the scientific campus of Messina University and at Alì are used for search activities. The teaching staff poses the following question to the attendee groups: *Are there concealed items in the ground of the suspect areas?* The team communicates that they can use: cords, wood pickets, evidence flags and markers, bags/boxes, marker pens, trowels, brushes, T-bar, metal detector, photographic camera, topographic maps, aerial imagery (remote sensing), GPS, laser scanner, GPR, and geophysical instruments for electrical resistivity and seismic tomography. The training in the search for buried targets starts in the simulated fields, and the first search step consists of a surface search for anomalies; attendee groups are arranged laterally in a human chain ("comb-like search") and proceed forward with a walkover. A laser scanner survey may be carried out by attendees with the assistance of the teaching staff to obtain a detailed and georeferenced 3D image of the surface for identifying possible surface anomalies. It is advisable to subdivide the search area with a square grid consisting of a cord. For each grid, the students may test the ground diggability index (the easiness of excavation) by using a steel T-bar and searching for metal items using the metal detector. Each evidenced anomaly is marked by a numbered flag and geolocalized using a GPS device. After the landscape analysis, the evidenced anomalies may be investigated by means geophysical methods, such as GPR and seismic imaging. Attendees carry out a GPR survey and seismic tomography (Figure 5) with the assistance of the teaching staff to verify the occurrence of ground geophysical anomalies in the previously identified surface anomalies. Both methods identify at the same point the occurrence of a geophysical anomaly. For forensic geologists, redundant results obtained from different methods are appreciable.

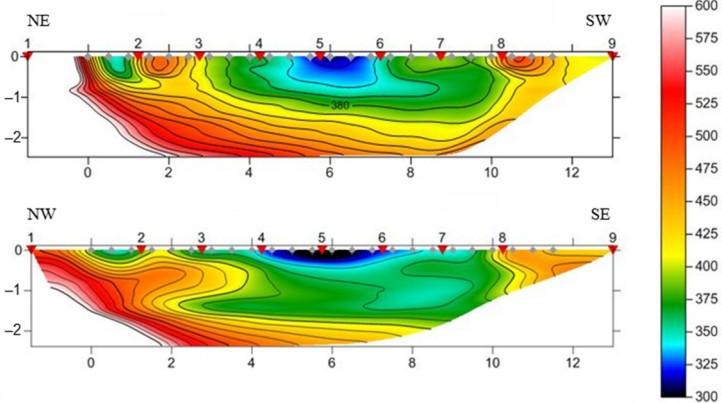

**Figure 5.** Two 2D Seismic Tomography sections show the simulated concealment tomb evidenced by anomalies characterized by low seismic wave velocity values. Source: own development.

After that, the geophysical investigation confirms the occurrence in the ground of anomalies previously evidenced on the surface; it is possible to stop the search activities to begin the forensic archaeological excavation (Figure 6) to recover possible human remains (simulated with mannequins) or other physical evidence. These delicate and sensitive activities are supervised by coroners, forensic anthropologists, and archaeologists. The excavation is carried out by attendees manually, using trowels and brushes; the excavated soil is collected for further analyses in the laboratory. Each evidence bag or box with the samples is closed, signed in a security bag or container, and stored in the laboratory.

Following conclusion of the field training, the teaching staff guides the attendee working groups to elaborate the report illustrating the answer to the question previously posed, based on the expert opinion to present to the judicial system (prosecutor, lawyers).

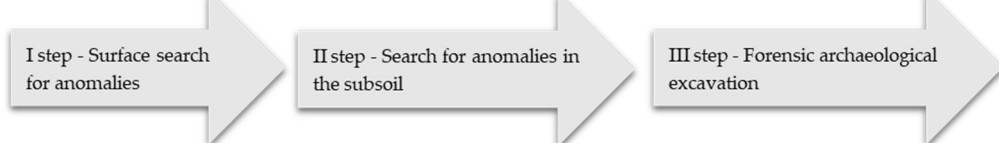

**Figure 6.** Workflow of the activities. Source: author.

3.1.3. Survey of a Site Hosting a Landfill

The survey activities in landfills are generally experienced in a closed landfill, whereas other inspections are carried out in the surrounding landfill. In both cases, the teaching staff poses the following question to the attendee groups: *Does environmental pollution of the groundwater associate with landfill leachate?* The team communicates to the students that they may also consider pre-existing chemical reports and that they can use: glass containers and bottles, freatimeter, bailers, portable pump and multiparametric probe, GPS, photographic camera, instruments for Electrical Resistivity Tomography, topographic maps, aerial imagery. The attendees are divided into two groups; the first one is devoted to the direct search for spring waters, and the second one to the indirect search for the ground water table. The first group verifies the existence of the ground water table using piezometers in the landfill and freshwater springs in the surrounding areas. All the individuated piezometers are localized using GPS, and the water table (or leachate) is measured by means of a freatimeter. Samples of liquid are taken from the piezometers using bailers and pumps, and water samples are taken from the spring using glass bottles. The color and odor of the liquid samples are examined. All the samples are photographed, closed, signed, and stored in a portable refrigerator before storing them in a laboratory refrigerator until chemists conduct chemical and physical analyses (Figure 7).

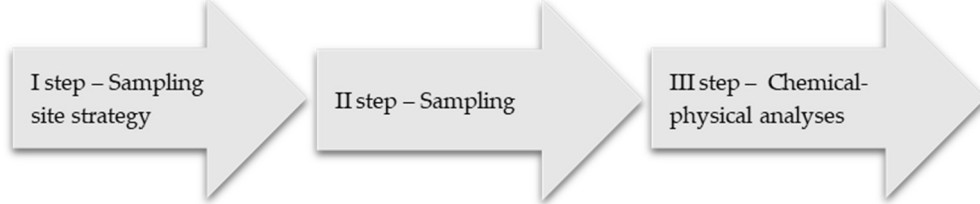

**Figure 7.** Workflow of the activities. Source: author.

The second group searches for indirect evidence of leachate within the landfill body and the water table in the ground out and under the landfill. To investigate the underground area, the attendees decide to apply geophysical methods by using 2D Electrical Resistivity Tomography and planning the localization of sections within the landfill and out of this to obtain electrical imaging (Figures 8 and 9).

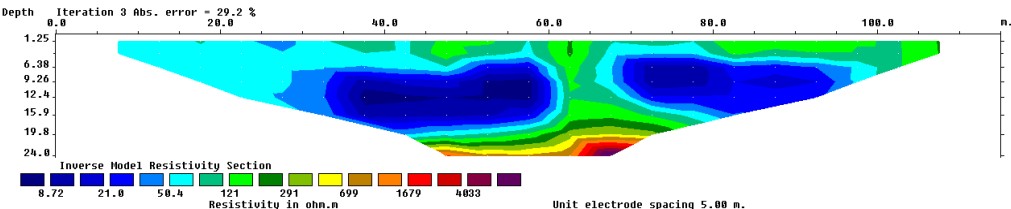

**Figure 8.** 2D Electrical Resistivity Tomography image showing the underground water distribution evidenced by the anomaly represented by low electrical resistivity values (blue color). Source: own development.

After conclusion of the field training, the teaching staff guides the attendee working groups to elaborate the report illustrating the answer to the question previously posed, based on the expert opinion to present to the judicial system (prosecutor, lawyers) (Figure 9).

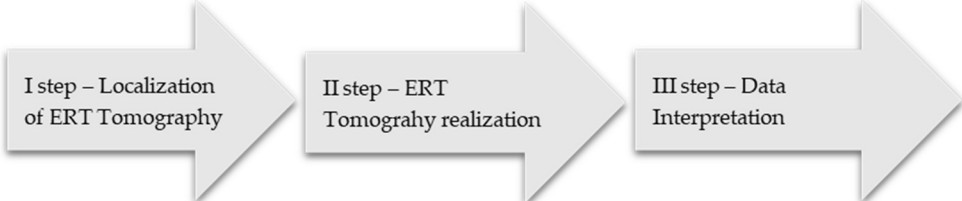

**Figure 9.** Workflow of the activities. Source: author.

*3.2. Laboratory Experience in Geological Physical Evidence Sampling and Preparation*

The teaching staff provides each attendee a dirty item with geological physical evidence and indicates the activities to carry out:

- "Crystallization" of the sample status (techniques to take forensic photographs);
- Sample collecting;
- Sample preparation;
- Grain size separation;
- Density fractions separation;
- Thin-section preparation;
- Smear slide preparation.

The staff communicates that during the experience, attendees may use: steel spatulas, spoons, brushes, orthodontic instruments, magnifying glass, stereomicroscope equipped with the photographic camera, oven, and sieves with 5 cm diameter, mechanical sieve shaker, equipment to utilize thin sections, scanner, and photographic camera. Each attendee disposes off a clean personal workstation on a numbered desktop where each one can pose the item upon a neat paper. Before the sampling, the attendee must carefully observe the features and the homogeneity degree of the traces/micro-traces. The item has to be carefully photographed according to different item faces, applying a graphic scale; the realization of a digital map (in GIS or other software) of the distribution areas of the trace (for localization and features) is very useful for the localization of the samples. When the geological physical evidence is homogenous and abundant, after this first observation and documentation phase, the traces may be sampled using spatulas, spoons, brushes, or orthodontic instruments based on the typology of the sole. It is recommended to preserve sub-sample of the sample for further use. When there are soil peds, these have to be held, marking the item's polarity for possible microstratigraphic analyses. Before carrying out any specific forensic examinations or treatment, it is essential to observe the general features of the samples and separate possible peculiar grains such as minerals and microfossils (inorganic component) or glass or brick fragments (anthropogenic component) in subsamples. This crucial analytical stage is realized through observation under the stereomicroscope. The organic component (roots, twigs, leaves, insects) may be removed by hand or tweezers. In contrast, organic matter may be removed by treating the geological physical evidence with specific procedures. The separation of the heavy minerals from the light ones may be realized by using heavy liquids [85] such as diiodomethane ($CH_2I_2$; Figure 10).

After the treatments, the sample may be air-dried or treated at 40 °C for at least 16 h in a laboratory oven until a constant weight is reached. The sample must be described and photographed under the stereomicroscope equipped with a photographic camera before the sieving separation. Sieving can be conducted using tiny sieves, being careful to use clean sieves. It may be helpful to put the sieves in water with methylene blue to avoid grain contamination. The separated fractions are stored in glass capsules or evidence plastic bags, carefully marked with an ID number or code. The attendees have to carry out an analogous procedure on the geological physical evidence collected from the crime scenes and alibi sites during the field training. A portion of each grain size fraction may be finally impregnated with resin (Figure 11) and mounted on a slide for inspection to obtain a thin section of the sample.

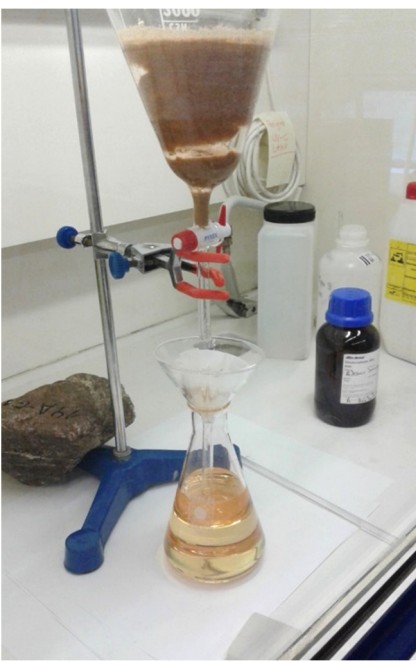

**Figure 10.** Separation of the heavy minerals from the light ones using diiodomethane ($CH_2I_2$), filters, and laboratory glassware under a chemical fume hood. Source: own development.

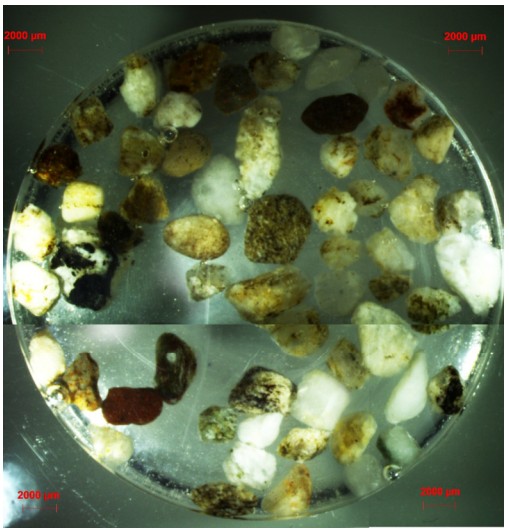

**Figure 11.** Sand fraction impregnated with resin before the cutting. Source: own development.

The smear slide method [86] may also be very useful for a rapid and low-cost preliminary observation, simply being prepared by posing a distilled water drop on a slide and "smearing" the geological physical evidence transferred on a toothpick. This technique for comparing samples represents a preliminary test that evaluates the degree of compatibility between micro-traces from an unknown sample and geological physical evidence from the crime scene (Figure 12).

The preparation of thin sections and smear slides (Figure 13) is necessary to conduct optical microscopy analyses for the sedimentological—mineralogical—microfossil characterizations.

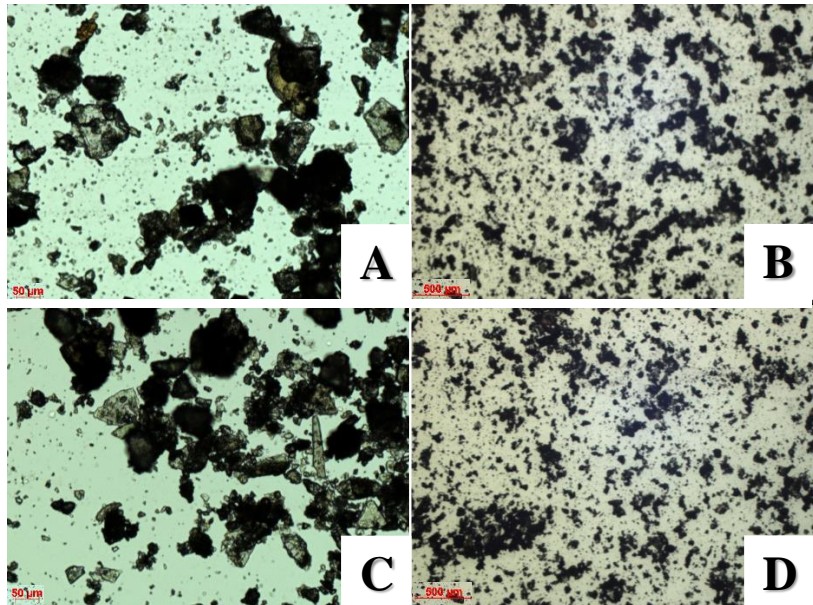

**Figure 12.** Microphotographs of forensic geological physical evidence in smear slides. (**A**) Unknown sample from the suspect's shoes (transmitted light). (**B**) Unknown sample from the suspect's shoes (reflected light). (**C**) Known sample from the crime scene (transmitted light). (**D**) Known sample from the crime scene (reflected light). Source: own development.

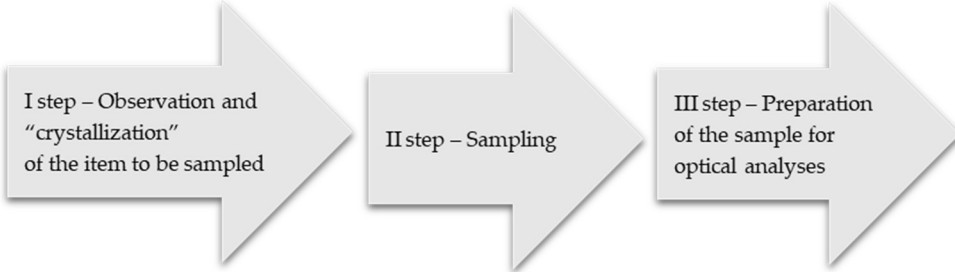

**Figure 13.** Workflow of the activities. Source: author.

### 3.3. Laboratory Experience on Geological Physical Evidence
### Color—Sedimentological—Mineralogical—Microfossil Analyses for Forensic Comparisons

Conventional laboratory techniques in forensic geology generally apply the following range of stereomicroscope, polarized light microscope, XRD, XRF, SEM-EDS, LD, FTIR, and RAMAN [11]. The teaching staff explains the activities involving geological physical evidence for color—sedimentological—mineralogical—microfossil comparisons to ascertain if unknown samples may be associated with the control samples or excluded from them (Figure 14).

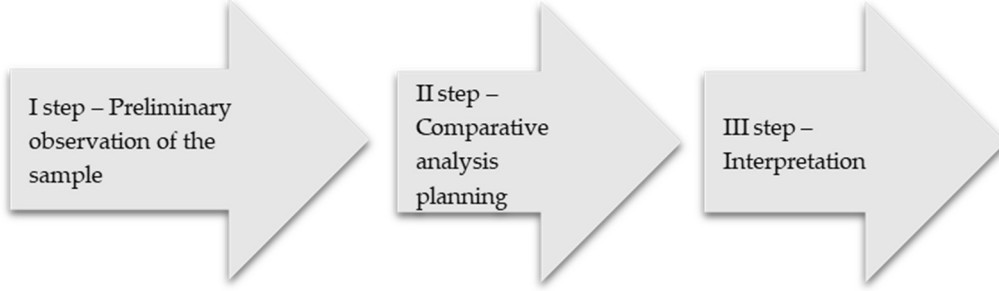

**Figure 14.** Workflow. Source: author.

Activities are subdivided into different attendee groups. Generally, the amount of earth materials available for forensic evidence analyses is often small and consequently, the methods used for forensic comparison vary in function of the sample size [27,87]. The staff communicates that during the experience, the attendees may use: spatulas, spoons, brushes, orthodontic instruments, photographic camera, agate mortar and pestle, Munsell soil charts, stereomicroscope, and petrographic microscopy equipped with the photographic camera and software for image analysis, instruments for powder X-ray diffractometry (XRD), scanning electron microscopy with energy dispersive systems (SEM-EDS), RAMAN spectroscopy and X-ray fluorescence (XRF).

Geological physical evidence initial analyses generally concern the color [88]. It is analyzed for each grain size fraction by applying different methods by working groups. It may be visually determined by comparison with standard charts of the color, defined by Hue, Value, and Chroma, observing the sample under the best conditions to natural light (preferably at 12:00 of the clock near a window exposed to the North). In addition, the attendees may scan all samples (contained in transparent plastic evidence bags) to obtain RGB colors. The color scanner is calibrated [89], and no correction is necessary for the plastic.

The texture analyses are carried out by working groups for various sand grain size fractions, both on untreated samples and thin sections of resin-impregnated geological physical evidence. The particle shape is analyzed by defining morphoscopic features and morphometric parameters (sphericity and roundness) on at least 100 grains under a workstation consisting of photographic camera-equipped microscopy provided by image analyses software. The grain sphericity may be calculated using different indexes (Figure 15A). Sphericity's and roundness's computations are also evaluated by using comparative charts. The luster (Figure 15B), the occurrence of coatings, and color are also assessed for each grain.

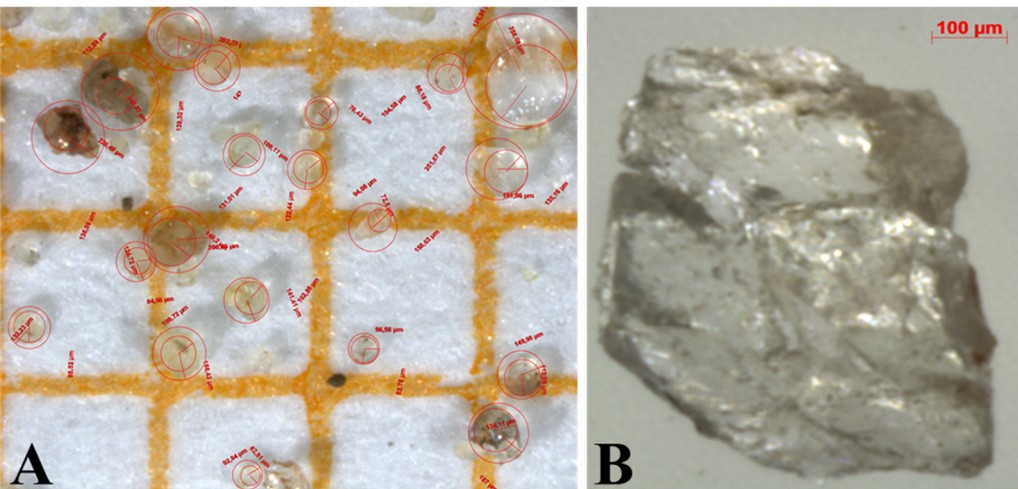

**Figure 15.** (**A**) Morphometric measures of the ray of the inscribed and circumscribed circle for calculating the Riley index of sand-size grains of quartz (observed under stereomicroscope; the orange grid is 1 mm × 1 mm). (**B**) Sand-size grain of hyaline quartz observed under a stereomicroscope. Source: own development.

After preliminary tests in smear slides (Figure 12), the mineralogical analyses are carried out by working groups in thin sections (Figure 16). Petrographic analyses allow students to estimate the relative abundances of the mineral composition of the samples.

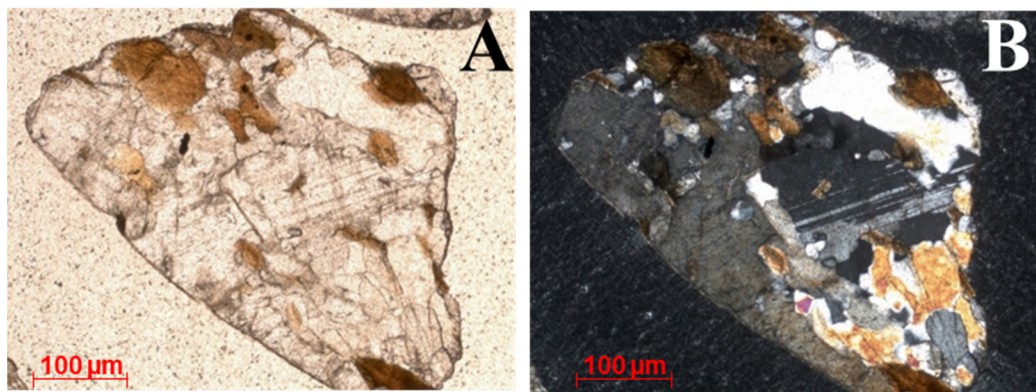

**Figure 16.** Microphotographs of a lithoclast of metamorphic rocks of the Aspromonte Unit in a thin section of an impregnated resin forensic soil from the Peloritani Mountains. The lithoclast is formed by a Variscan gneiss composed of quartz + plagioclase + biotite mineral paragenesis. (**A**) Microphotograph took in plane-polarized light. (**B**) Microphotograph took in crossed polarized light. Source: author.

The qualitative mineralogical composition is confirmed by XRD [38] and SEM-EDS analyses [35] made by the attendees. XRD is conducted on bulk samples after grinding using an agate mortar and pestle. Samples are packed in sample holders and scanned from 2° to 70° in a diffractometer (Figure 17). Software is used to display diffractograms and to identify mineral composition.

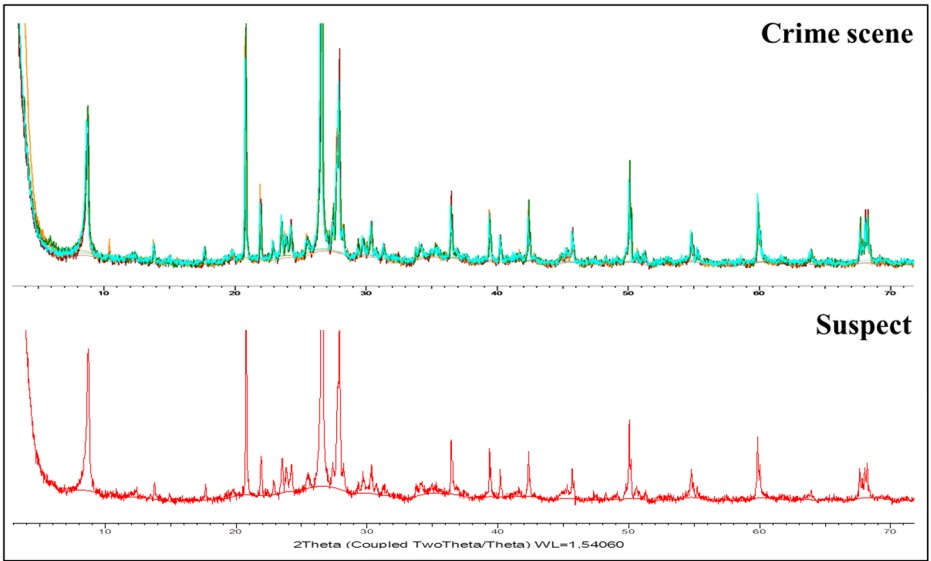

**Figure 17.** Diffractograms of forensic geological physical evidence (red color line: unknown sample from the suspect's shoes; green color lines: known samples from crime scene). Bulk samples in both cases are siliciclastic and composed of the same minerals (quartz, plagioclase, biotite, and muscovite). Source: author.

Attendees conducted SEM-EDS analyses both on polished sections of impregnated resin samples and stubs of inorganic/organic particles to characterize surface topography and composition (Figure 18). SEM-EDS may be replaced by automated instrumental systems, such as the QuemSCAN [90], which is very useful for identifying and quantifying the mineral composition of samples limiting the operator's influence on the interpretation.

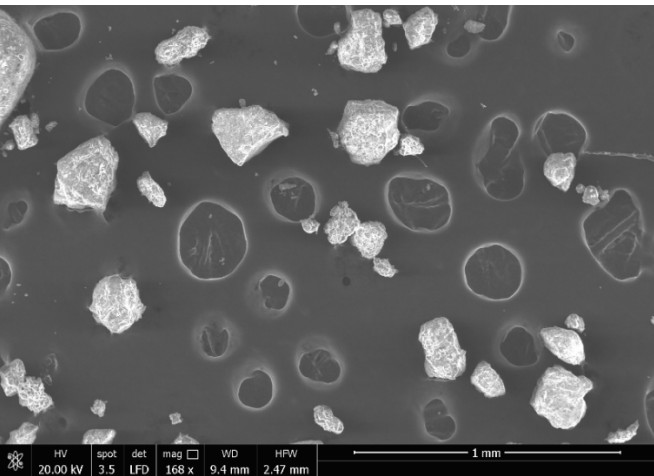

**Figure 18.** Microphotograph of quartz grains seen under SEM. The hexagonal prismatic habitus of the quartz grains may be recognized. Source: author.

Unusual anthropogenic components, such as brick, and mortar [91], paper, plastic and glass fragments, or paint slices found in the geological physical evidence, maybe also characterized using X-ray fluorescence XRF and RAMAN spectroscopy. Attendees analyze some paint slices by XRF (Figure 19).

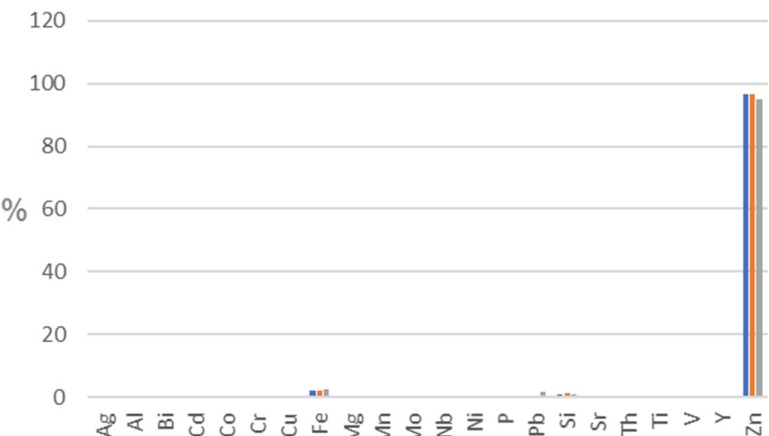

**Figure 19.** Three anthropogenic slices of paint in a forensic soil sample were analyzed by XRF, indicating a zinc-rich composition. Source: Own development.

Microfossils found in the geological physical evidence may be identified under a stereomicroscope (Figure 20) and or SEM-EDS. The most common microfossil occurring in forensic soils are foraminifers, but rare microfossils may be present. They may assume a strong probatory significance for comparative analyses in this latter case. Attendees are invited to observe geological physical evidence samples under the stereomicroscope to search for microfossils.

In Figure 20, it is possible to observe that a rare microfossil recognized by the attendees and classifiable as conodont after SEM-EDS analyses confirmed a phosphatic composition. This taxon was found in forensic soil collected by the author and whose provenance may be related to rare outcrops of Silurian-Devonian rocks of the Calabria-Peloritani Arc [92–95].

After the conclusion of the field and laboratory training, the teaching staff guides the attendee working groups to participate in an inclusive and final experiential exercise for the elaboration of a final report illustrating all the answers to the questions previously posed, based on the expert opinion to present to the judicial system (prosecutor, lawyers). Considering the theoretical issues and experiences developed during the summer school, attendees have to consider with critical thinking all results to answer the questions. The

teaching staff suggests that the interpretation results may be better synthesized in synoptic tables such as that in Table 1 related to the comparative analyses.

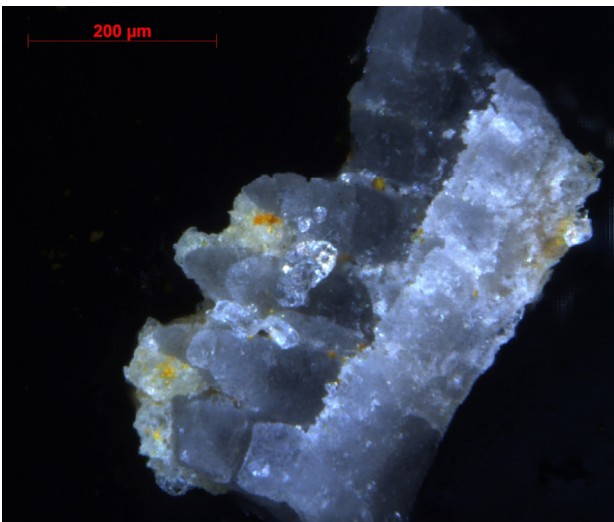

**Figure 20.** Stereoscopic microphotograph showing a peculiar bioclast of conodont (photograph was taken using the Z-stack procedure). Source: author.

**Table 1.** Synoptic table showing the expert opinion on the compatibility/similarity degree of geological physical evidence collected on the suspect (unknown samples) with geological physical, geochemical and mineralogical evidence sampled in two different zones (areas 1 = crime scene; 2 = alibi area). The results of the comparative analyses between the traces/micro-traces collected on the suspect and the soils from area 1 show that a high degree of compatibility is appreciable. In contrast, the soil from area 2 may be excluded. The source: own development.

| PARAMETERS | SUSPECT AREA 1 (KNOWN SAMPLES) | SUSPECT AREA 2 (KNOWN SAMPLES) |
|---|---|---|
| COLOUR | HIGH DEGREE | LOW DEGREE |
| MINERALOGY | HIGH DEGREE | HIGH DEGREE |
| GRANULOMETRY | HIGH DEGREE | LOW DEGREE |
| MORPHOLOGY | HIGH DEGREE | LOW DEGREE |
| PECULIAR PARTICLE | Biotite Clasts | / |

## 4. Discussion

Training and fieldwork, experiential, interactive, collaborative, and outdoor learning may be the teaching and learning key strategies for academic education in forensic geology. The presented approach in this paper is based on the flipped classroom method for teaching and learning forensic geology and represents a modern and helpful means to develop inclusive and independent learning and acquire critical thinking skills on the value of geological physical evidence. This method is contra posed to the most traditional and old teacher-centered approach where the students are considered to be so called "empty vessels" that passively absorb information [96]. This non-traditional and experiential learning approach, used at the Messina University since 2014 both in the field and in the laboratory, proves to be an opportunity for teaching staff worldwide that desire to develop academic training activities in forensic geology and for geology undergraduates/postgraduates interested in building a professional career in forensic geology. The model may represent successful tools for multitasking teaching and learning strategies necessary to further develop forensic geology, especially in Italy, to encourage the inclusion of forensic geologists within police enforcement and improve training activities for experts of the prosecutor and the defense lawyers. A review of the main advantages and challenges of the flipped classroom method suggested that it may successfully improve the students' learning

performance, satisfaction, and engagement [64]. Notwithstanding, it may be that some scholars cannot correctly organize their time for the out-of-class learning curriculum [97]; the academic pedagogical experience matured by the author suggests that this risk for undergraduates/graduates is moderate. Future research of the Author will be devoted in formulating and interpreting questionnaires on the offered education approach to submit to attendees of the next forensic geology course editions.

**Funding:** This research was funded by institutional funds, University of Messina.

**Institutional Review Board Statement:** Not applicable.

**Informed Consent Statement:** Informed consent was obtained from all subjects involved in the study.

**Data Availability Statement:** The data presented in this study are available upon reasonable request to the corresponding author.

**Acknowledgments:** The author would like to thank the hundreds of attendees and police officers who have participated since 2014 in the forensic geology educational activities promoted by Messina University. A special acknowledgment for their support goes to the teaching staff (Orazio Barbagallo, Giorgio Cecchini, Francesco Crea, Salvatore Gurgone, Caterina Ingoglia, Vittorio Longo, Antonio Marchese, Sebastiano Monaco, Corrado Rizzo, Massimiliano Silvestro, Luca Sineo, Eliana Torre, and many others among police officers and magistrates). All the author's actions to contrast the environmental crimes are in memory of the police officer Tiziano Granata. The anonymous reviewers are kindly acknowledged for their constructive reviews, which led to significant improvements of the original manuscript.

**Conflicts of Interest:** The author declares no conflict of interest.

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
