# Peer review of "Advances in Flipped Classrooms for Teaching and Learning Forensic Geology"

_education, doi:10.3390/educsci12060403_

Round 1

Reviewer 1 Report

The manuscript is well written and no major edits are needed.  However, I do feel that "Figure 3" could be more focused on students participating a particular exercise.  A group photo composite does little to add scientific merit to the manuscript. 

Author Response

Dear Reviewer 1,

Thank you very much for your revision and observations especially relating to figure 3.

I tried to focus better on students participating exercises in the field and in the laboratory, and I prepared another version of Figure 3 (Figure 3 NEW) in attached.

Best regards

Reviewer 2 Report

This paper provides a strong foundation for the publication of a journal paper on “the approach to training forensic geology students and law enforcement offices (police and lawyers etc)”.

However, many places throughout the paper required extensive clarification by rewriting text. This reviewer has substantially edited the PDF (see attached). It is critical the author or authors carefully review the numerous edits made using the PDF editing tools.  

In order to get this review submitted in a timely fashion this reviewer has not taken the time to extract all these edits and comments from the manuscript and place them in this review document.  I  suggest that the author examine the comments and changes to the edited PDF manuscript with track changes and accept the suggested changes or explain why not and do a further revision of the manuscript, also incorporating the overall suggested changes.

In regard to the suitability of this paper to be published in the Journal of Education Sciences, this reviewer believes it is suitable if appropriate revisions are made in response to this reviewer’s comments and suggestions outlined especially made directly on the PDF and in the margins of the PDF document (attached).

This reviewer believes that it is a good paper that could become a much better read if the author or authors would consider addressing the critical errors of terminology and wording used throughout the current manuscript, which can easily be corrected.

Author Response

Dear reviewer 2,

I carefully reviewed and accepted all the edits made (except "is" at line 11 because included in another correction; peddles at line 437, because I wanted to say peds; and " at line 649).

I followed all the corrections and suggestions.

I'm sorry but all revisions made to the manuscript were not marked up using the“Track Changes” function, but with green color.

I have to kindly acknowledge you for your constructive reviews, which led to significant improvements of the original manuscript.

I strongly appreciated all the useful corrections and suggestions.

Thank you very much.

Best regards
